# *House of Dextra*: CROSS-EMBODIED CO-DESIGN FOR DEXTEROUS HANDS

**Kehlani Fay**[1]   **Darin Djapri**[1]   **Anya Zorin**[1]   **James Clinton**[2]   **Ali El Lahib**[1]

**Hao Su**[1]   **Michael T. Tolley**[1]   **Sha Yi**[1]   **Xiaolong Wang**[1]

[1]University of California, San Diego      [2]University of California, Santa Barbara

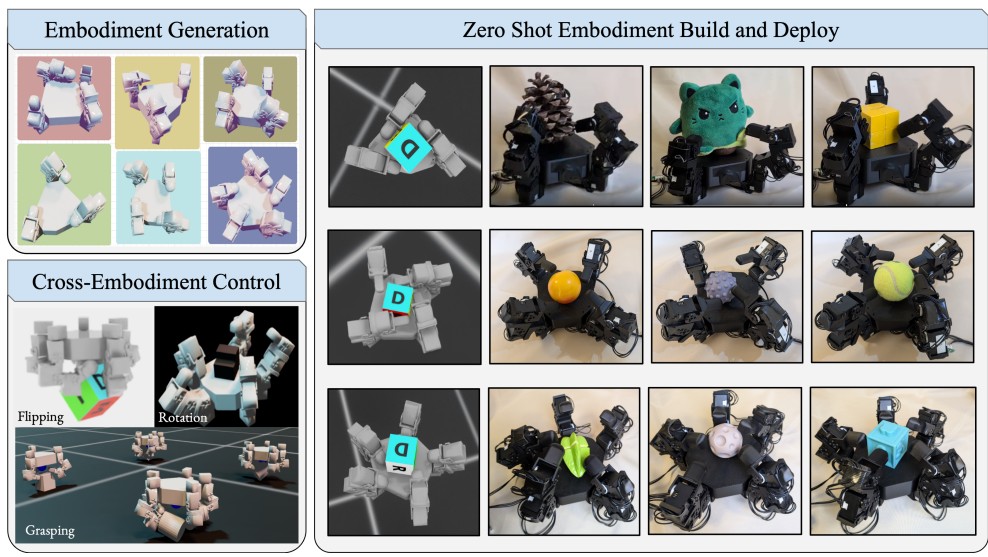

Figure 1: *House of Dextra* is a cross-embodied co-design framework for dexterous hands which jointly optimizes control and hardware design. Using this framework, we generate four co-designed dexterous robot hands in simulation and transfer them zero-shot to the real world.

## ABSTRACT

Dexterous manipulation is limited by both control and design, without consensus as to what makes manipulators best for performing dexterous tasks. This raises a fundamental challenge: how should we design and control robot manipulators that are optimized for dexterity? We present a co-design framework that learns task-specific hand morphology and complementary dexterous control policies. The framework supports 1) an expansive morphology search space including joint, finger, and palm generation, 2) scalable evaluation across the wide design space via morphology-conditioned cross-embodied control, and 3) real-world fabrication with accessible components. We evaluate the approach across multiple dexterous tasks, including in-hand rotation with simulation and real deployment. Our framework enables an end-to-end pipeline that can design, train, fabricate, and deploy a new robotic hand in under 24 hours. The full framework is open-sourced and available on our website.

## 1 INTRODUCTION

Endowing robots with human-level dexterity critically requires both the development of fine control (Wang et al., 2025; Qi et al., 2023; Zeng et al., 2019) and dexterous hardware design (Zorin

et al., 2025; Shaw et al., 2023; Unitree Robotics; Allegro Hand; Christoph et al., 2025). Traditionally, robot manipulation has been developed by decoupling mechanical design from control, with reinforcement learning and control frameworks developed on fixed hardware architectures. This separation can limit dexterity - as optimal control policies may be fundamentally constrained by the morphology of the hand, degrees of freedom, and sensing capabilities (Yin et al., 2025; Calandra et al., 2018). Co-design aims to overcome this limitation by simultaneously adapting design and control (Cheney et al., 2013; Pathak et al., 2019; Bai et al., 2025).

Additionally, previous works (Yuan et al., 2021; Lu et al., 2025; Carlone & Pinciroli, 2019; Wang et al., 2023a; Cheney et al., 2013) have often been deployed only in simulation. The sim-to-real gap remains for many works due to over simplifications in hardware manufacturing constraints Aljalbout et al. (2025), simplifications in simulated contacts to real world dynamics Huang et al. (2023), and modeling of physical properties Zhao et al. (2020b). To overcome this, we create a modular hardware platform with realistic components in simulation, enabling designs for sim-to-real.

*House of Dextra* creates a **cross-embodied co-design framework** enabling quick and efficient evaluation of co-designed robot hands and allowing design to sim-to-real in under 24 hours. **Modular Robot Hand**: We create a robot hand platform with modular mechanical and electrical structure. This allows exploration of variable kinematics, degrees of freedom, and morphology. We show this by deploying four different robot hand embodiments sim-to-real. **Accurate Robot Hand Generation**: We create modular and physically realistic grammars and rule structures that generate robot hands that can be built and deployed sim-to-real. **Design Analysis:** We expand co-design beyond primitive parameters for manipulation and identify parameters of impact for future dexterity research.

The result is a cross-embodied co-design framework with efficient evaluation and generation of robot hands. We demonstrate this approach sim-to-real for 18 novel objects with in-hand rotation across multiple generated designs. Results can be found on our website: HouseOfDextra.

## 2 PROBLEM SETTING AND NOTATIONS

We jointly optimize robot embodiment and control so that structure and behavior are aligned. Let $\mathcal{G}$ denote the morphology space, $\Pi$ the policy space, and $\mathcal{D} \subseteq \mathcal{G}$ the set of evaluated designs. Our goal is to co-design a hand morphology $G \in \mathcal{G}$ and policy $\pi \in \Pi$ that maximize task reward.

**Morphology Representation.** In our implementation, each hand is represented as a fixed-topology attributed graph $G = (V, E, X_v)$, with one palm node and five finger-slot nodes ($|V| = 6$). Edges encode only connectivity, $E = \{(0, i)\}_{i=1}^{5}$, i.e., a star graph from palm to fingers. Each finger node has attributes $x_v \in X_v$, including servo count, grammar codes, fingertip type, terminal/active flags, and finger index (with group context).

**Graph Neural Networks.** To account for isomorphic graph structure, we encode each morphology graph $G$ with a message-passing GNN $f_\phi$. Given $(V, E, X_v)$, $f_\phi$ performs $K$ rounds of message passing to produce a fixed-dimensional embedding, $y(G) = f_\phi(G) \in \mathbb{R}^d$. This embedding is used by the design value network during graph heuristic search to evaluate and rank candidate morphologies.

**Morphology-Conditioned Control Policy.** We model dexterous manipulation for a fixed morphology $G$ as an MDP $\mathcal{M}_G = (\mathcal{S}, \mathcal{A}, T, R, \gamma)$, where the dynamics depend on the embodiment. The policy is conditioned on a one-hot morphology vector $m(G) \in \{0, 1\}^{d_m}$ with $d_m = F_{\max}(1 + 2L_{\max})$, where $F_{\max} = 5$ and $L_{\max} = 3$. The state is defined as $s = (q, p_o, m(G))$, where $q \in \mathbb{R}^{|E|}$ are robot joint states and $p_o \in \mathbb{R}^6$ is the object pose.

The action space $\mathcal{A}$ consists of joint position commands. Since not all actuators are present in all morphologies, we apply an action mask $M(G) \in \{0, 1\}^{|\mathcal{A}|}$ to the raw policy output: $a_t = \pi_\theta(s_t) \odot M(G)$, where $\odot$ denotes element-wise multiplication. This allows a shared policy to operate across different embodiments without producing invalid actions. The reward $R(s_t, a_t)$ is task-specific and depends on object state, joint effort, object proximity to the hand, and related task

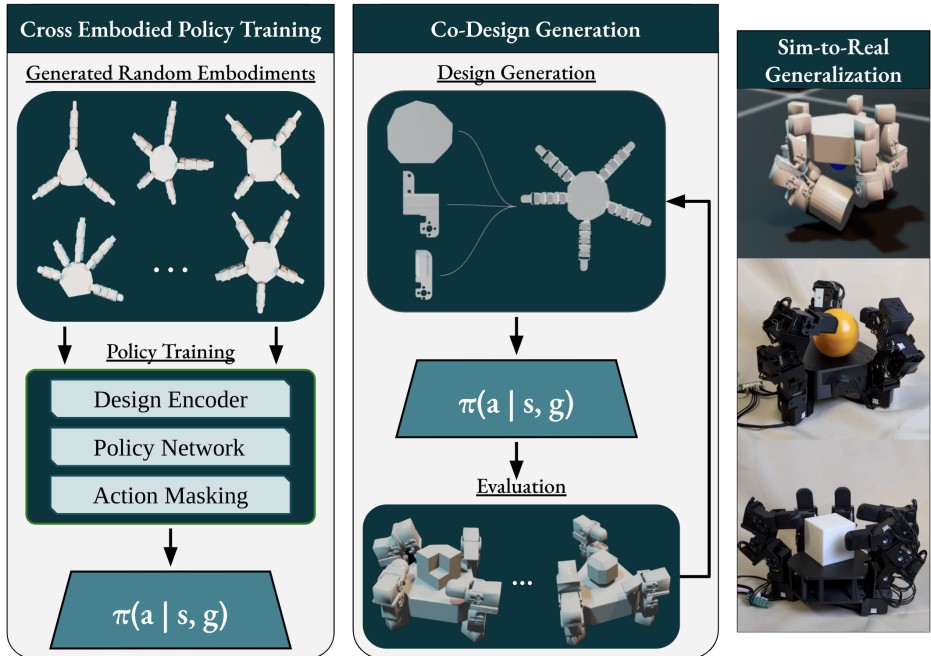

Figure 2: *Method overview.* **Stage 1:** We randomly sample embodiments across morphologies and degrees of freedom and pre-train a morphology-conditioned policy across embodiments. **Stage 2:** Designs are generated using modular grammars and evaluated in simulation by the cross-embodied policy. **Stage 3:** The best designs and fine-tuned control are manufactured and deployed on real hardware across unseen objects.

terms. For a fixed morphology $G$, the policy objective is

$$J(\pi, G) = \mathbb{E}_{\tau \sim \pi, \mathcal{M}_G}\left[\sum_{t=0}^{T} \gamma^t R(s_t, a_t)\right], \tag{1}$$

where $\tau = (s_0, a_0, s_1, \dots)$ denotes a trajectory in $\mathcal{M}_G$.

**Co-Design as Bi-Level Optimization.** The co-design problem jointly optimizes morphology and control:

$$(G^*, \pi^*) = \arg\max_{G \in \mathcal{G},\, \pi \in \Pi} J(\pi, G). \tag{2}$$

Equivalently, this can be written as the bi-level problem

$$G^* = \arg\max_{G \in \mathcal{G}} J(\pi_G^*, G), \tag{3}$$

$$\pi_G^* = \arg\max_{\pi \in \Pi} J(\pi, G). \tag{4}$$

The inner loop optimizes control for a fixed morphology, while the outer loop searches for the morphology with the best achievable task performance. Because this joint space is large, we first train a cross-embodiment base policy $\tilde{\pi}_\theta$ over sampled designs $G \sim p(G)$, then use it to efficiently evaluate and fine-tune generated designs during co-design.

## 3 METHOD

Our approach enables sim-to-real robot hand co-design through cross-embodiment learning across embodiment families. An overview of our method is shown in Figure 2. We first generate a set of diverse embodiments by randomly sampling design rules. Rather than learning separate policies for each hand design, we group morphologies into families based on shared kinematic structures, asymmetric or symmetric, and variable finger counts. This enables efficient generalization across fingertip

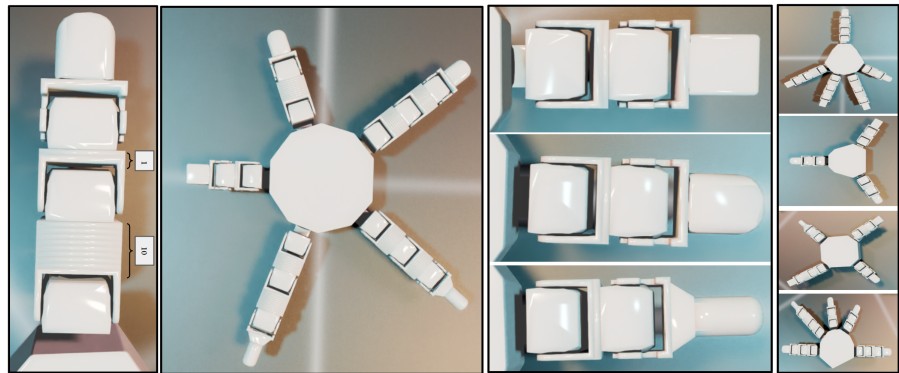

Figure 3: Left to Right. 1) Variable finger length of 1-10 using flat modular stacks. 2) Example embodiment with varying degrees of freedom with fingers of 3 or 4 servo motors. 3) Fingertip Variations 4) Finger number and example embodiment variations.

types, degrees of freedom, link lengths, palm geometries, and finger placements within each family. To make the control policy aware of the morphology, while considering the isomorphism of design, we condition the control policy on encoded morphology. Our physical grammar system reflects real hardware constraints through pre-computed collision geometries, joint limits, and manufacturable component specifications, ensuring designs transfer effectively to physical systems.

**Components.** The framework consists of four main components:

$$
\begin{array}{llll}
\textit{Generation} & G \sim p_{grammar}(r, c) & \triangleright \text{Procedural constraint hand generation} & (5) \\
\textit{Design encoder} & y = f_\phi(G) & \triangleright \text{Graph embed for design-value optimization} & (6) \\
\textit{Design value network} & V_{design}(y(G)) & \triangleright \text{Learned design eval for search guidance} & (7) \\
\textit{Cross-embodiment policy} & \pi_\theta(a \mid s, m(G)) & \triangleright \text{One-hot morphology conditioned control} & (8)
\end{array}
$$

Here, $y(G) = f_\phi(G)$ is the graph embedding for design evaluation, $m(G) \in \{0, 1\}^{d_m}$ is the one-hot morphology vector for policy conditioning, and $s, a$ denote state and action variables. The grammar generator encodes palm geometry, finger count $n_f \in \{3, 4, 5\}$, per-finger joint configurations $j_i \in \{2, 3\}$, segment-scale grammar codes $c_1, c_2 \in \{1, \dots, 10\}$, and fingertip types using real modular sub-components with realistic collision geometries and manufacturing constraints (Appendix Fig. 2). During co-design, the system iteratively generates candidate morphologies, evaluates them with the pre-trained cross-embodiment policy, and selects designs for manufacturing.

### 3.1 MODEL OBJECTIVE

Design evaluation leverages cross-embodiment to assess morphology performance. The design value network learns from observed design-performance pairs:

$$
\mathcal{L}_{design} = \mathbb{E}_{d \sim \mathcal{D}_{evaluated}} \left[ (V_{design}(y(G_d)) - \mathcal{T}(d))^2 \right] \tag{9}
$$

where $d$ indexes an evaluated design instance and $\mathcal{T}(d)$ represents the maximum observed task performance for that design under the cross-embodiment policy. The search explores $\mathcal{G}$ through Graph Heuristic Search (Zhao et al., 2020a), maintaining a lookup table $\mathcal{T} : \mathcal{D} \to \mathbb{R}$ that maps evaluated designs to empirical scores and guides value network training. At each iteration, $K$ candidate designs are generated through sequential grammar expansion. For each partial design, legal next rule applications are scored by the design value network, where $v_i$ is the predicted value of the $i$th expansion. These scores are perturbed with Gumbel noise, $v_i' = v_i + \tau g_i$, where $g_i \sim \text{Gumbel}(0, 1)$ and $\tau$ is a noise scale controlling exploration strength. The next expansion is selected by maximizing $v_i'$. This keeps the search value-guided while still exploring alternative design choices. The lookup table is then updated via $\mathcal{T}(d) = \max(\mathcal{T}(d), \hat{J}(G_d))$ for newly evaluated designs.

## 3.2 ARCHITECTURE AND TRAINING

**Hand Generation for Cross-Embodiment Pre-training.** We develop a parametric hand generator that constructs diverse, physically-valid morphologies through a two-stage process. First, the generator samples palm layouts by placing finger bases on a variable circular radius under configurable placement modes (symmetric, asymmetric, or anthropomorphic), discretizing the circle into slots with minimum angular separation enforced via rejection sampling. A convex palm mesh is constructed by computing a 2D convex hull of grip points around each motor position and extruding it to a specified thickness. Second, each finger's internal morphology is independently sampled from a structured design space controlling: (1) the number of actuated joints (2–3 servos beyond the base), (2) two discrete grammar codes $c_1, c_2 \in \{1, \dots, 10\}$ that determine proximal and distal segment scales, and (3) fingertip geometry (standard, wedged, rounded, or thin variants). This produces thousands of hand variants by sampling from uniform distributions across grammar parameters with fully randomized designs. Each generated hand includes pre-computed collision geometries via CoACD convex decomposition for all modular components, realistic joint limits derived from CAD specifications, and consistent kinematic representations.

**Cross-Embodiment Policy Pre-training.** We group generated hands into kinematic families based on structural similarity—symmetric radial configurations, anthropomorphic arrangements, and variable finger counts—enabling targeted policy learning within each family. The cross-embodiment policy uses PPO enhanced with morphology-conditioning:

$$\mathcal{L}_{policy} = -\mathbb{E}_{(s,G)} \left[ \min \left( \frac{\pi_\theta(a|s, m(G))}{\pi_{old}(a|s, m(G))} \hat{A}_t, \text{clip} \left( \frac{\pi_\theta(a|s, m(G))}{\pi_{old}(a|s, m(G))}, 1 - \epsilon, 1 + \epsilon \right) \hat{A}_t \right) \right] \quad (10)$$

where $m(G)$ represents the one-hot morphology encoding provided to the policy. Each family trains separately on 2000-8000 hand variants, with the policy learning unified control strategies that generalize across finger counts, joint configurations, and geometric variations within the family. This morphology conditioning enables the policy to adapt its control strategy based on the specific kinematic structure and available degrees of freedom.

**Graph Heuristic Search Algorithm.** Following cross-embodiment pre-training, we employ Graph Heuristic Search to optimize hand designs for specific tasks. At each iteration, the search generates $K$ candidate morphologies, evaluates them in parallel using the pre-trained cross-embodiment policy, and updates both the lookup table $\mathcal{T} : \mathcal{D} \to \mathbb{R}$ and the design value network $V_{design}(y)$ based on observed task performance. The lookup table stores the best observed score for each evaluated design and provides supervision for value network training. This allows the search to focus computation on promising regions of the morphological space while maintaining diversity through grammar-based design generation.

**Design Optimization** The search begins from an initial design with a randomly configured palm layout and randomized finger bases, marking fingers as non-terminal nodes requiring expansion. At each iteration, the algorithm generates candidates through sequential finger-by-finger expansion: for each non-terminal finger, it evaluates all valid parameter choices by constructing successor designs and scoring them with the GNN, which predicts performance from graph representations augmented with Gumbel noise (noise_scale $\approx 0.4$) for exploration. The highest-scoring successor is selected, repeating until all fingers are finalized into complete designs. These candidates are evaluated in parallel simulation across multiple hand groups, yielding task-specific performance scores. The lookup table is updated with each complete design's score and its partial ancestors' scores (enabling credit assignment during construction), using effective keys that account for morphological equivalences (finger permutations in symmetric layouts, thumb slot variations in anthropomorphic layouts). The GNN is trained via supervised regression on this growing dataset, minimizing squared error between predicted and observed scores with L2 regularization and gradient clipping. This updated value network guides subsequent candidate generation with refined predictions, while an annealing epsilon schedule ($0.4 \to 0.05$) shifts from exploration to exploitation, and path-level tabu tracking prevents revisiting identical construction sequences within each batch, ensuring sustained diversity.

**Sim-to-Real Transfer.** The best-ranked design from search is fine-tuned with domain randomization over actuator characteristics, contact and friction parameters, and object poses to improve robustness (Andrychowicz et al., 2020; Tobin et al., 2017). For deployment, we train the controller

Table 1: Comparison of best designs for continuous in-hand rotation in rad/s. Baselines search across 2000 designs with stronger rewards implemented for manipulation (Appendix 9.5). For the LEAP hand we use the Leap-Reorientation environment and train their default policy code and object (1 object). We train another policy on our randomized object dataset without object state (blind) as these are representative of our real world task.

| Method | Run Time (hours) | Cont. Ang. Velocity (rad/s) |
|---|---|---|
| **Ours** | 6.48 | 3.3 |
| Ours w/o fine tuning | 5.18 | 1.85 |
| Ours w/ MPPI | 20.0 | 0.62 |
| Leap, Single Cube w/ Vision | 2.0 | 0.47 |
| RoboGrammar | 23.0 | 0.26 |
| Monte Carlo | 15.2 | 0.20 |
| Blind Leap Hand | 2.0 | 0.0 |

as a *blind* policy, i.e., removing object-state inputs, to match onboard sensing and reduce sim-to-real mismatch. There is no camera or tactile feedback. The corresponding cross-embodiment policy is fine-tuned from the family checkpoint using stateless observations. Manufacturing the real robot hand follows the same programmatic grammar used in simulation: the best selected design graph is directly converted to modular hardware specifications with 3D-printed components and precisely matched actuator constraints. A mounting interface is added to the palm, each joint and links are directly 3D printed, and then assembled with Dynamixel actuators. After tuning PID on hardware, the blind policy is deployed on the physical system.

## 4 EXPERIMENTS

We evaluate our co-design framework on three dexterous manipulation tasks, including one with simulation and real-world deployment for in-hand rotation using generated designs. Our experiments demonstrate that cross-embodiment learning paired with modular hardware graph grammars enables efficient and generalizable co-design for manipulation without sacrificing the runtime or dexterity commonly lost when using traditional co-design control methods. We show successful performance across multiple tasks and demonstrate effective sim-to-real transfer.

**Environments.** We test the co-design framework across three manipulation tasks, including sim-to-real in-hand rotation. These tasks comprise in-hand rotation, grasping on a table top, and object flipping about the z axis on a table top with fifteen randomized objects. We run 50 iterations with 40 designs at each iterations. Each evaluation cycle tests candidate designs in parallel across 2048 randomized simulation environments, including randomized initial joint states, physical parameters, and object type. See 6 for setup.

**Baselines.** We compare our co-design framework against the following baselines for in-hand rotation: (1) RoboGrammar (Zhao et al., 2020a), a procedural generation algorithm using graph-based design rules and local search optimization for robot morphology design; (2) Monte Carlo Tree Search, a tree-based exploration method that systematically searches the design space by building a search tree and using random rollouts for evaluation; (3) Our method with MPPI control; (4) The LEAP hand with vision trained on a single object trained with PPO Shaw et al. (2023); (5) A blind LEAP hand with our randomized objects for comparison to standard hardware.

Compared to these baselines, our method achieves finer control and is able to successfully complete manipulation tasks. Robogrammar achieves a best design rotation speed of 0.26 rad/s and a time to fall of 4.63 seconds for the in-hand rotation task. By contrast, our best designs without fine tuning achieves a 1.85 rad/s in simulation with no time to fall over the evaluation time of three minutes. Monte Carlo performs worse than the graph heuristic search, with a time to fall of 2.71 seconds. We find reinforcement learning is substantially better suited for manipulation co-design tasks as it can better overcome sparse rewards, long horizon tasks, and achieve complex finger gaits needed for true dexterity.

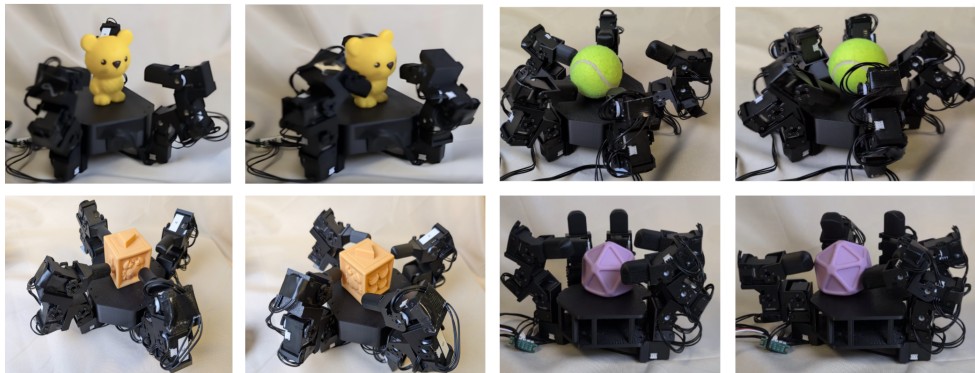

Figure 4: **Top Left:** Three finger co-design hand, best design found by algorithm. **Top Right:** Five finger symmetric co-design hand. **Bottom Left:** Four finger co-design hand with thin fingertips. **Bottom Right:** Five finger anthropomorphic baseline.

**Sim-to-Real Transfer.** To demonstrate our framework's real-world applicability, we conduct zero-shot blind deployment from simulation to reality. We use the best found design to deploy a blind policy with our random rotation objects within our domain randomization constraints. Three co-design embodiments are shown to show hardware tractability of our modular setup and show design tradeoffs. One baseline anthropomorphic hand is also deployed sim-to-real for comparison. We deploy the policy zero-shot, closed loop with proprioception without state feedback of the object type/position. Only encoder information of servo motors and the morphology encoding are used as the observation.

## 5    RESULTS: ANALYSIS OF GENERATED ROBOT HANDS

### 5.1    REAL WORLD DEPLOYMENT

To show our method is able to be built and transferred to the real world, we tested seventeen unseen objects with variable friction, compliance, and shape. A blind policy was trained in simulation on a subset of high performing designs found by rotation simulated environment. We compare three co-design hands, including the best found optimal 3-fingered hand, to an anthropomorphic baseline. We find robot hands with radial symmetry greatly outperform anthropomorphic robot hands on blind in-hand rotation. We choose these designs to show the impact of morphology on task success and the ability to accurately predict task success ranking in real.

Table 2: End-to-end pipeline timing for generating, manufacturing, and deploying one robot hand sim-to-real.

| Component | Time (hours) |
| --- | --- |
| 3D Printing | 12.0 |
| Algorithm | 6.48 |
| Assemble | 0.8 |
| Sim-to-Real | 2.0 |

From simulation results, the anthropomorphic robot hand trained across the randomized objects achieves a 0.73 rad/s in simulation. The three fingered robot hand achieves 1.85 rad/s without fine tuning. The robot hands were deployed with the morphology-conditioned cross-embodiment policy. The inputs are the morphology's encoding and the joint state positions only, with no state information, while the robot hands ran across objects until either reaching a minute of continuous rotation or failure.

We show the time used to rotate a given object 360 degrees, and a subset of the results is shown in Table 3. Failure to rotate within one minute is denoted as a dash. The three fingered robot hand executed grasps which were able to generalize more effectively to unseen objects. The anthropomorphic and four finger robot hands were able to blindly rotate a small subset of objects - three out of seventeen tested objects, as predicted to be lower performing by our algorithm. Objects often became stuck between finger gaits. Across the same seventeen objects, the three fingered robot hand generalized substantially better generalization. Across all seventeen objects, only two failed with one

Table 3: Four designs are shown in order of simulation success, showing the accuracy of our algorithm sim-to-real. Task performance (seconds) across hand types and objects. Dash indicates not achieved.

| Hand type | Pink polygon | Tennis ball | Rubik's cube | Green block | Purple ball | Star fruit | Pink disk | Peach can | Yel. block |
|---|---|---|---|---|---|---|---|---|---|
| **3-finger, optimum** | 9 | 13 | 9 | 9 | 5 | – | 15 | 10 | 6 |
| 5 Finger | – | 12 | – | 18 | 8 | – | 14 | 18 | – |
| 4 Finger | – | 10 | – | – | 9 | – | – | – | – |
| Anthropomorphic | – | 25 | – | – | 13 | – | – | – | – |

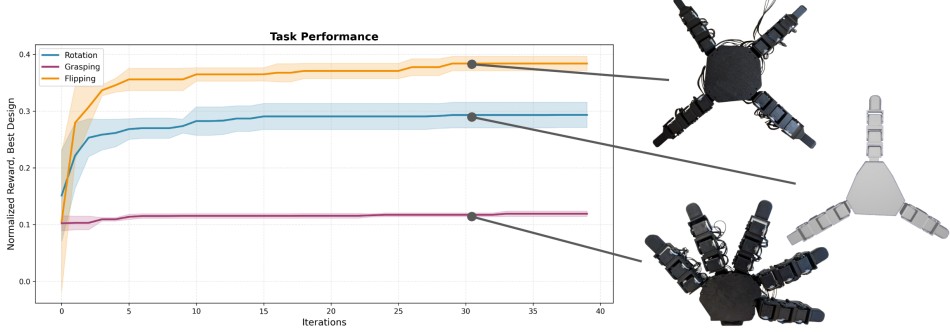

Figure 5: Normalized reward of best design for flipping, in-hand rotation, and grasping.

two small between the robots bling gait to reach. The three fingered morphology generalized across irregular objects including pine cones, pentagons, and soft objects. The gait of the three fingered robot hand is three altering diagonals, allowing the object to be rotated with little displacement. See videos for details.

## 5.2    SIMULATION TASK : DESIGN RESULTS

In simulation, we deploy across three tasks including grasping, in hand object rotation, and object flipping. **Grasping**: Grasping is lifting the randomized object to the palm with a stable grasp and fixed wrist. Across randomized objects, thinner fingertips were primarily picked with 54% of designs having this feature. All best designs had full degrees of freedom per finger (4 servo motors) and five fingers of symmetric and anthropomorphic layouts. These designs complete a successful grasp without dropping in under a half second. **Rotation**: In hand rotation had the highest standard deviation, with 25th percentile hands having less than 25 degrees of rotation per second and the top 75th percentile hands more than double (67 deg/sec). Best designs used three fingers achieving - without fine tuning - 1.85 rad/s across objects and 3.3 rad/s with fine tuning. Standard fingertips worked best with 64% of designs using this attribute. Thinner fingertips resulted in unstable rotation. **Flipping**: Flipping is composed of rotating the object around the z axis underhand and requires forward rotation with passing the object back to a start position for additional rotation. Best designs (58% of all generated designs) used wedged or thin fingertips on one side to lift the object easily and standard fingertips on the other side of the palm to have controlled reset with four and five finger symmetric hands. Across all tasks, full degrees of freedom (4 servos per finger) consistently performed asymmetric designs, including those with thumb layouts. As this is the hardest task due to a fixed wrist and variable object heights, the best designs achieve flipping in less than 6 seconds.

## 6    WHY MORPHOLOGY?

We conduct a parameter analysis for robotic hand manipulation by systematically sampling across 12 critical hardware categories in the Isaac-Reorient-Cube-Leap task on a LEAP hand (Shaw et al., 2023). Our sampling approach used uniform grid sampling with 100 samples per parameter across their physically meaningful ranges (Appendix Fig.  14-16), covering three parameter categories: physics properties (static/dynamic friction, restitution), actuator characteristics (stiffness, damping,

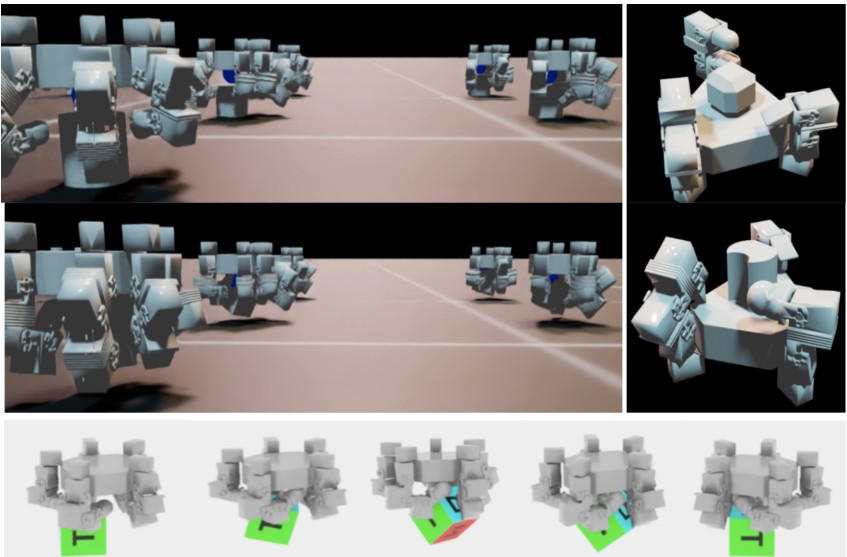

Figure 6: **Left:** Grasp task with over a minute hold time and quick grasps.**Bottom:**Flipping task's best found design with rotation on z-axis with a fixed wrist. **Right:** In hand rotation setup.

effort limits, velocity limits), and morphological scaling factors (palm width, finger length, mass scaling). Each parameter class was trained using PPO, yielding a dataset of 1,200 training runs.

Our analysis (Fig. 5 and Appendix Grid Sampling) reveals four parameters with particularly strong effects on manipulation performance: finger body length scale showed the strongest positive correlation (r=0.748), while palm width scale exhibited a strong negative correlation (r=-0.729), suggesting that wider palms impede dexterous rotation. Of these, morphology parameters had the highest positive impact. Additionally, two motor control characteristics, damping (r=-0.476) and dynamic friction (r=-0.459), showed moderate negative correlations with performance. This provides a strong argument for co-design since the top four parameters can be optimized through control and morphology. Material properties were found to be of lower impact. A non-linear relationship is observed through polynomial curve fitting indicate that optimal manipulation performance occurs within specific parameter ranges rather than at extremes. Morphology and control having the most pronounced impact on task success compared to material or contact properties.

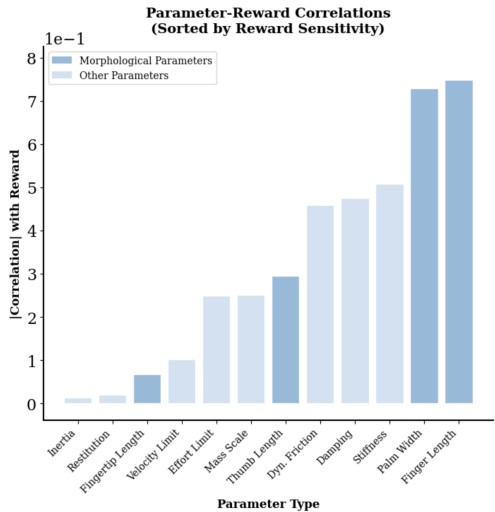

Figure 7: Correlation of physical parameters to improving rotation from Bayesian Sampling on a LEAP hand. Parameters of highest impact are morphology and control parameters.

## 7 RELATED WORKS

**Co-Design of morphology and control policy.**
Traditional design approaches first fix hardware architecture, then optimize control, co-design treats morphology and policy as a joint space that often outperforms separate optimization (Ha et al., 2016; Schaff et al., 2019; Matthews et al., 2023). However, the morphology-policy joint optimization is high-dimensional (Chen et al., 2021). Previous works focused on optimizing the design only (Ha

et al., 2021; Xu et al., 2024; Kodnongbua et al., 2023) based on a fixed control policy, or co-design with a limited design space (Luck et al., 2020; Islam et al., 2024; Yi et al., 2025; Dong et al., 2024). Most co-design work has remained in simulation due to challenges in sim-to-real transfer of both control policy and manufacturable designs (Xu et al., 2021a; Xiong et al., 2023; Dong et al., 2023a). While most works focused on locomotion (Yuan et al., 2021; Wang et al., 2023c; Dong et al., 2023b), manipulation presents unique challenges due to the large hardware design space (Piazza et al., 2019), fine control for contact-rich dynamics (Qi et al., 2023; Yuan et al., 2020; Zakka et al., 2023). Evolutionary algorithms scale poorly to complex morphologies(Lipson & Pollack, 2000; Cheney et al., 2013). Programmatic spaces, as in Zhao et al. (2020a), add compositional rules for valid topologies, improving sample efficiency and structured exploration. We combine programmatic design with cross-embodiment learning to efficiently search the joint space without sacrificing fine motor control.

**Cross-Embodiment Learning** Learning policies that generalize across different robot morphologies addresses the sample efficiency limitations of training separate models for each embodiment (Doshi et al., 2024; Gupta et al., 2021; O'Neill et al., 2024). Current approaches leverage shared neural network architectures with morphology-conditioned inputs (Huang et al., 2020; Attarian et al., 2023; Furuta et al., 2022), function approximators that encode embodiment parameters (Zhao et al., 2020a), morphological pretraining (Li et al., 2024; Strgar & Kriegman, 2025), and meta-learning frameworks for few-shot adaptation to new morphologies (Nagabandi et al., 2018; Wang et al., 2023b). Graph neural networks have shown particular effectiveness for variable-topology robots by learning permutation-invariant representations over robot joints and links, enabling zero-shot transfer to previously unseen kinematic structures (Wang et al., 2018; Liu et al., 2023; Lu et al., 2025). However, transfer performance degrades significantly with large morphological differences, particularly in contact-rich manipulation tasks where small design variations can drastically affect policy performance (Doshi et al., 2024).

**Sim-to-Real Transfer** Distribution shift between simulated training and real-world deployment remains challenging - especially for co-designed systems (Chen et al., 2021; Schaff et al., 2023). The reality gap stems from complex physical dynamics, control noise, and manufacturing tolerances that compound when both morphology and control are optimized in simulation (Peng et al., 2018; Muratore et al., 2022) Current solutions include system identification techniques that adapt learned policies to real hardware through online fine-tuning (Du et al., 2021). While recent work combines reinforcement learning with differentiable optimization for locomotion (Xu et al., 2021b), co-design for sim-to-real and dexterous manipulation remains largely unexplored.

## 8 LESSONS LEARNED AND LIMITATIONS

Through our experiments on cross-embodied co-design for three dexterous manipulation tasks, we find the following: **1) Cross-embodiment enables tractable co-design scaling.** Cross-embodiment evaluation provides a practical approach to assess designs for complex manipulation without sacrificing fine control for computational speed, making co-design optimization feasible for real-world deployment. **2) Realistic grammar-based design enables zero-shot transfer.** Grounding design grammars in physical components allows for seamless sim-to-real transfer without additional fine-tuning, as the modular structure maintains consistent physical properties across domains. **3) Task-specific optimization can outperform anthropomorphic design.** Our results demonstrate the success of morphologies optimized for specific manipulation tasks (flipping and rotation). Further research in general non-anthropomorphic hands can be explored. **4) Morphology dominates the design space.** Among all design parameters, morphology has the largest impact on performance, suggesting that shape and structure should be prioritized over other design considerations.

**Limitations.** While our framework demonstrates effectiveness across multiple manipulation tasks, several key bottlenecks remain. Our current approach only recomputes palm geometry while keeping other components modular and pre-defined, limiting full exploration of the morphological design space. Additionally, individual designs remain optimized for specific tasks rather than generalizing across multiple manipulation skills—incorporating multi-task design averaging or weighting mechanisms represents a critical direction for future work. Finally, our framework focuses primarily on morphological optimization while neglecting other critical parameters such as compliance, material properties, and actuation schemes that could further improve manipulation performance.

**Acknowledgment** Kehlani Fay is supported by NSF Graduate Research Fellowship under Grant No. DGE-2038238 and DGE-2545911. This project was supported, in part, by gifts from Amazon, Meta, and Qualcomm. A kind thank you to Lars Paulsen for help rendering mechanical rendering.

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

Figure 8: Example rollouts of MPPI/CEM control on the LEAP hand for in-hand rotation. The planners generate jitter-heavy, throwing-like motions that produce brief rotation but fail to maintain stable manipulation over long horizons.

# 9 APPENDIX

## 9.1 CONTROL TRADEOFFS

### 9.1.1 MPPI FAILURE ON SUFFICIENT HARDWARE

To ensure simple and fast control optimization was not sufficient for long horizon manipulation tasks, an additional experiment was conducted using MPPI to rotate a cube using a LEAP robot hand which has previously proven to perform in hand palm rotation using PPO Shaw et al. (2023). We ran MPPI and Cross Entropy Method on 5 initial grasp positions all within a few millimeters of touching the cube of 0,04 m size and 0,15 kg. Over the tested iterations of 60, 120, and 500 prediction horizon and 40 sample trajectories, no solution found was able to withhold a time to fall longer than 3 seconds. We tested across three different objects including the cube mentioned prior, a ball of 0,04 m in size and 0,1 kg, and a smaller cube of 0,2 kg. Of these solutions, many were jitter heavy and prone to throwing behaviors to achieve initial short horizon rotation rewards. No successful rotation was completed, especially as horizon time was increased. Altering these controllers to handle sparse rewards and long horizon tasks remain an open area of research.

### 9.1.2 CROSS EMBODIED PPO V. INDIVIDUAL PPO PERFORMANCE

**Runtime Analysis.** To evaluate the computational efficiency of our cross-embodied approach, we compared it against training individual PPO policies for each morphology. Training a single design using PPO required over 26 hours on average. Given these constraints, we were only able to evaluate 20 designs, with an average evaluation time of 47 minutes per design. In contrast, our cross-embodied evaluation framework evaluated 2,000 designs in just 5.18 hours—a **400× speedup** in runtime while enabling evaluation of 1,800 additional designs in one-fifth of the time.

**Fair Comparison Protocol.** To ensure a fair comparison with the PPO baseline, we reduced the training epochs from 1,000 (used for the cross-embodied policy) to 350 epochs per morphology. This reduction reflects the empirically observed convergence time for individual hand policies and provides a more equitable runtime comparison.

**Performance Results.** Despite the dramatic computational savings, our cross-embodied approach demonstrates no noticeable performance degradation compared to individually trained PPO policies. Moreover, the cross-embodied policy significantly outperforms single-morphology PPO training in several key metrics. For example, when trained from scratch on a single five-finger symmetric hand, PPO achieves 0.56 rad/sec rotation. The cross-embodied policy, deployed with *no fine-tuning* on the same morphology, achieves a **65% performance improvement**. Additionally, the cross-embodied policy exhibits greater robustness, with fewer failures across objects, while single-morphology PPO policies frequently fail to achieve meaningful rotation on the same training object set. Detailed results are provided in the accompanying video.

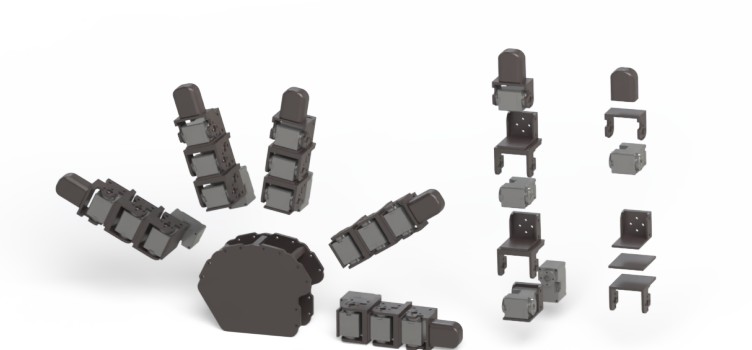

Figure 9: Exploded view of modular robot hand, anthropomorphic baseline.

## 9.2 HARDWARE OVERVIEW

We use Dynamixel XL330-M288-T servo motors to control each joint. The system is fully 3D printed from carbon fiber filament for strength. All components were manually designed, then separated into modular pieces with collision meshes computed. Each hand is approximately 7.5 inches in height and 6 inches in width. For generation, convex palms of length 0.08 meters were computed around generated finger placements. Palms are 1.5 inches thick with mounting in the back for directly mounting two rotational servo motors which control 90-degree rotation side to side of each finger. Additionally, each horizontally placed linkage is composed of a base mount, motor-controlled 90-degree flexion (forward and back), and a hinge to which modular topology components are attached.

For alterable components, we swap each offset linkage to be 0.1 inches, allowing for lengthening of each finger. Similarly, the topmost linkage is one of four selected fingertips of choice combined with the topmost hinge. This allows us to combine and print these components at once for quick and easy assembly. For electrical control, we use a U2D2 power hub with the expansion pack. This allows up to 20 motors to be controlled at once. Many cheap, direct drive, or open-sourced anthropomorphic hands do not have five fingers (Allegro, LEAP). Creating our own hardware setup allows a more accurate comparison of anthropomorphic designs while ensuring electrical and mechanical alterations. We chose to create our own hardware in order to ensure accurate transfer from simulation to real-world application, allow modular components, and ensure the electrical system could change dynamically with designs.

Additional designs can be easily printed for each component, by remounting to the servo horn. This allows for new fingertips, finger shapes, palm shapes, or material properties to be explored in the future. Co-design involves both significant hardware and software design. A hardware and manufacturing guide is presented on our website. By providing a hardware platform that can be easily adapted, we hope to make co-design research more accessible to the broader research community.

Our robot hand baseline is 10% smaller in length than LEAP and has five fingers instead of four. It is also 25% smaller than Allegro. We maintain the same degrees of freedom per finger as LEAP with a base rotation joint and 3 finger hinges for flexion.

## 9.3 REAL WORLD SETUP

We test 17 unseen objects using the blind proprioception only policy. No object state is given to the policy, only morphology encoding and joint state positions for each finger. No vision or object state is given to the system, requiring the robot hands to predict object state from resistance of the combination of encoder resistance values. These objects include soft and rigid boxes, weighted and light objects, irregular shapes, variable friction and surface finishes, and textures. No vision is used at deployment.

In our experiments, four robot hands are deployed from simulation to real. The anthropomorphic robot hand was picked from random from our generated dataset to give a comparison to typical five finger, anthropomorphic morphology. The other three robot hands were deployed as follows: **3 Finger Robot Hand:** the optimal robot hand found by our co-design search, **5 Finger Robot Hand:** A well performing hand in simulation to show ability to accurately predict design ranking order sim-to-real, **4 Finger Robot Hand:** a moderate to low ranked robot hand by our framework to show poor performance of thin fingertips, and **Anthropomorphic Hand:** to compare to anthropomorphic morphology as a lower ranked rotation design and ability to predict design ranking sim-to-real.

In simulation, the predicted performance for each robot hand is 1.85 rad/sec across objects for the 3 finger robot hand. For symmetric, five finger design, predicted performance is 1.57 rad/sec. The four finger robot hand varied highly on objects due to fingertips causing greater object displacement, but averaged about 0.78 rad/sec in simulation. The anthropomorphic robot hand by contrast receives a 0.62 rad/sec sim-to-real. These results show the ability of co-design to evaluate designs successfully in simulation and predict their real world performance.

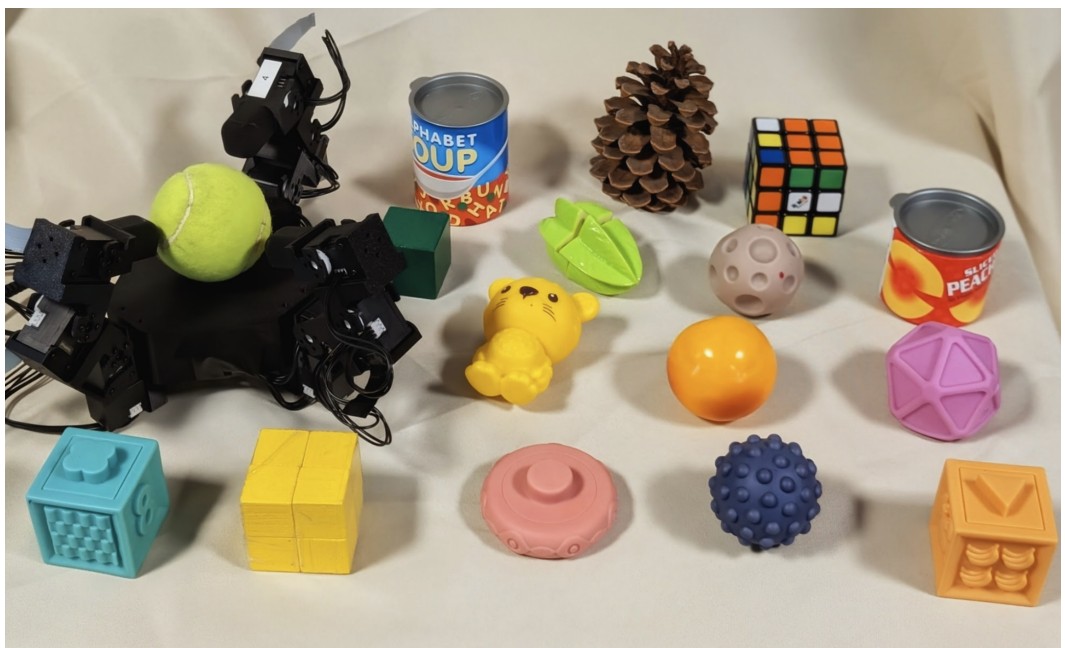

In real, the objects picked ranged from 8g to over 100g with varying textures, mass, compliance, and center of mass. Our three finger robot hand was able to generalize significantly better across objects, including rotating a real world pinecone with adaptive gaits to object position. This also best seen by being able to rate the tennis ball for over 10 minutes without failure while other designs cause greater displacement and are unable to predict object state sufficiently for blind rotation to complete rotation for this duration.

### 9.4 ENVIRONMENTS

We created three simulation environments including object rotation, grasping, and flipping. The results of these can be seen in our video. Prior works in co-design for manipulation have almost exclusively focused on grasping. However, we find this task had the least sensitivity to design for our setup, compared to in-hand rotation and flipping. We think this tradeoff is likely due to using rigid bodies and that in-hand rotation and object flipping are dynamic tasks with much more sensitive constraints for fine dexterity.

### 9.5 STRONGER BASELINES

RoboGrammar uses PyBullet, and we reimplement our environments in PyBullet to test across. Without these alterations, the method does not work at all for manipulation. We alter RoboGrammer's generation rules and reward function as follows:

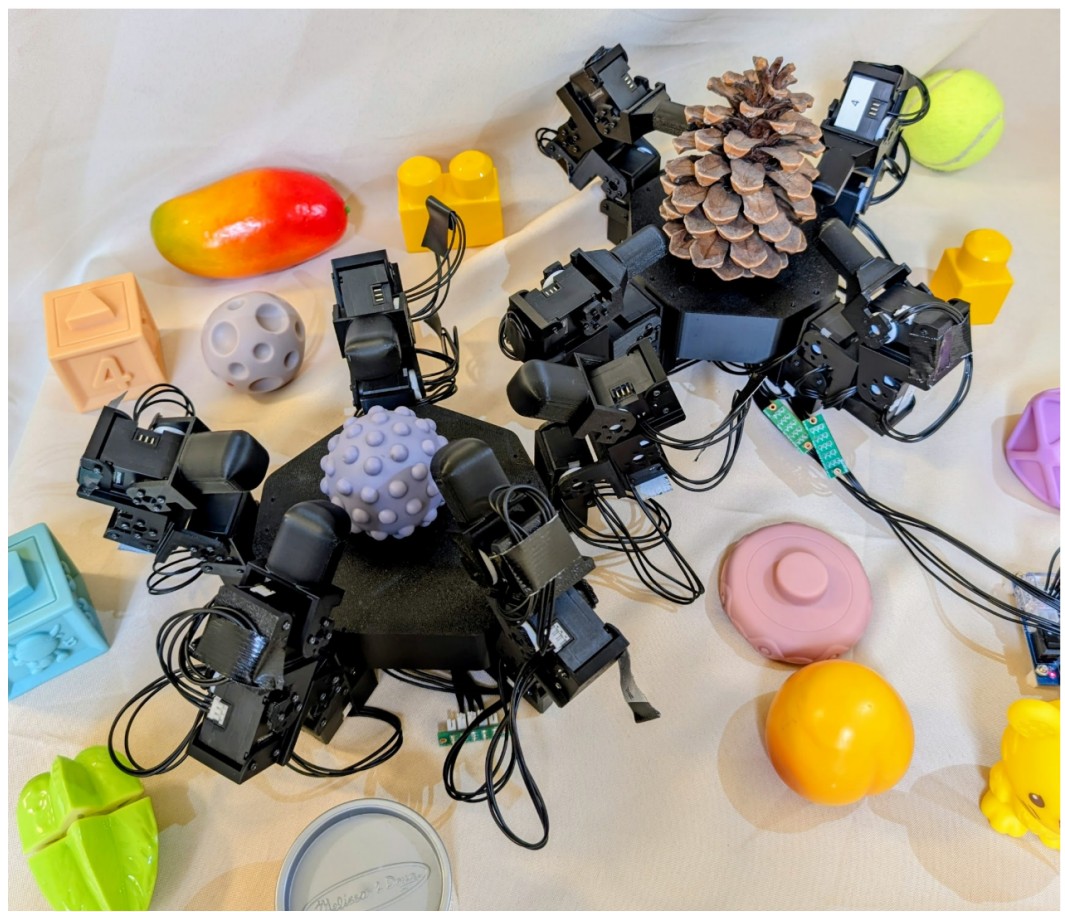

Figure 10: Two sampled co-design hands including the 5 finger with standard fingertips and four finger symmetric hand with thin fingertips are deployed sim-to-real. Both are made of the same modular components, but with varying finger lengths, palm shape, finger number, and fingertips.

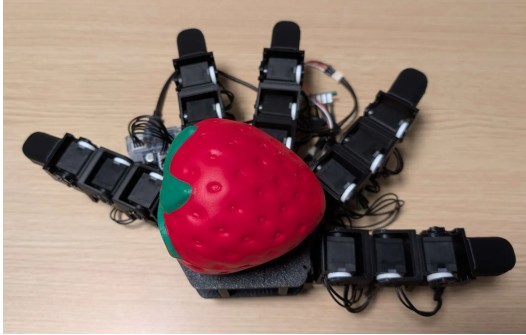

Figure 11: Modular anthropomorphic robot hand baseline.

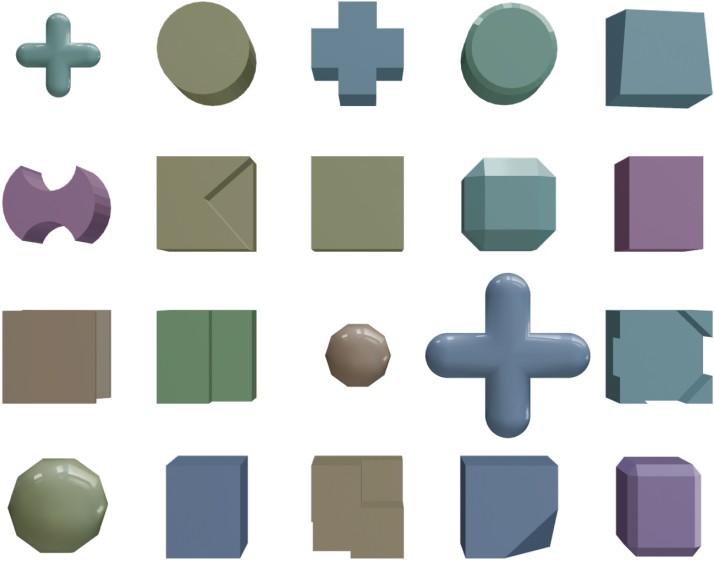

Figure 12: The object dataset used for evaluating designs across tasks in simulation. Objects were domain randomized for scale sim-to-real with size variance of 0.7 to 1.3 times size. Mass, position noise, and additional factors were randomized.

# Robot Design Rules

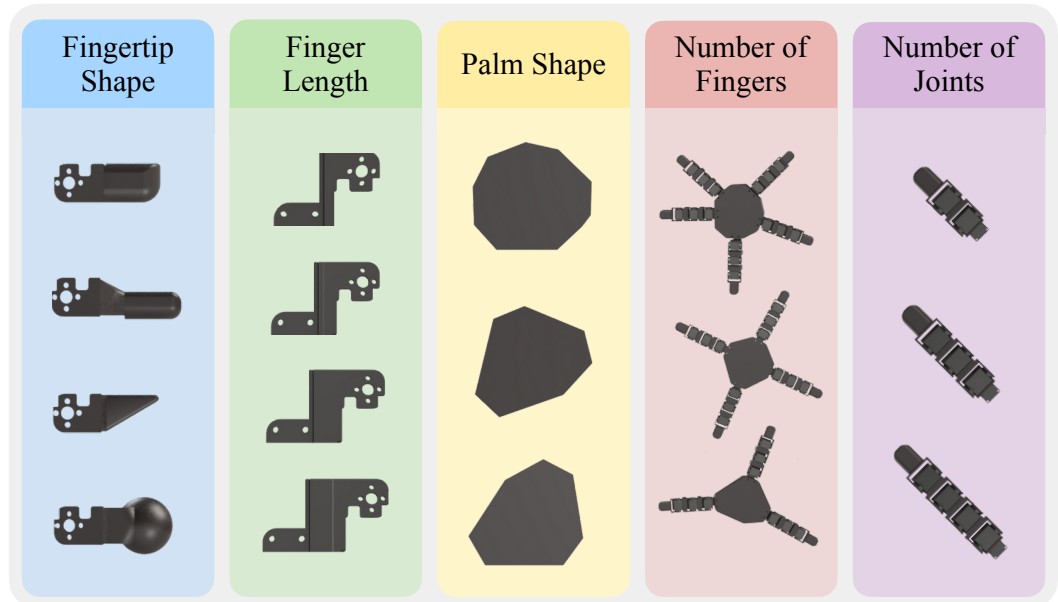

Figure 13: Five parameters are optimized including fingertip type from selection shapes inspired from LEAP Shaw et al. (2023), finger length of 1-10 linkage extensions for each joint, finger placement and palm shape as determined by a convex hull of finger placement, number fingers 3-5, and number of joints 3-5 joints including a default base rotation joint for side to side movement.

Table 4: Simulation Physics and Actuator Configuration

| Parameter | Value |
|---|---|
| **Rigid Body Properties** | |
| Static friction | 0.2–1.5 (randomized) |
| Dynamic friction | 0.4–1.5 (randomized) |
| Restitution | 0.0–0.5 (randomized) |
| Object density | 1000 kg/m³ |
| Bounce threshold velocity | 0.2 m/s |
| **Actuator Settings** | |
| Effort limit | 0.35 Nm |
| Velocity limit | 7.2 rad/s |
| Stiffness | 2.5–3.1 Nm/rad (randomized) |
| Damping | 0.05–0.15 Nms/rad (randomized) |
| Friction | 0.01 |

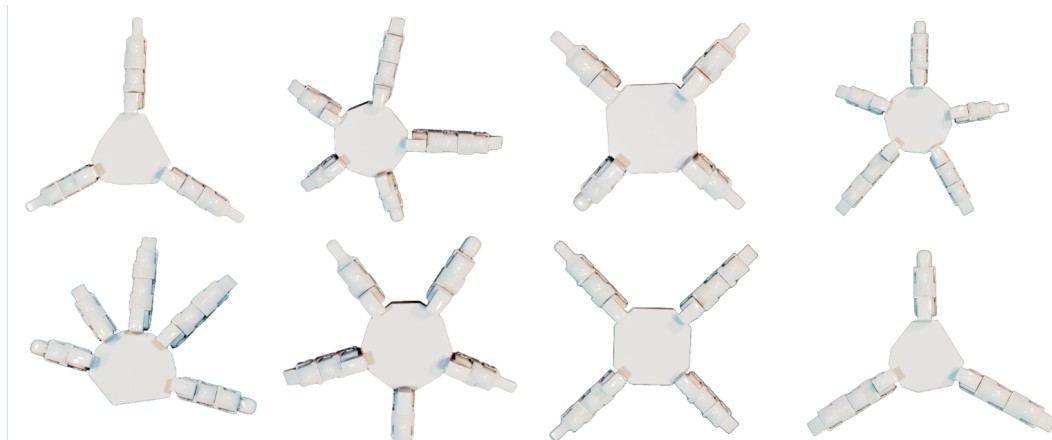

Figure 14: Random examples of generated robot hands.

The initial body node was changed to a palm shape, represented by a short and wide cylinder. The body node included 5 potential starting points, and the grammar was changed to include the option of adding limbs to these body nodes or terminating them. The angles of the joints were changed to face upwards to allow for in-hand manipulation in place of locomotion. Otherwise, the grammar rules were unchanged. The reward function consisted of a reward for the angular velocity of the object in one direction, a penalty for no motion of the object, and a penalty for object displacement from its initial position. This results in a throwing action to spin the cube.

We originally did not account for consistent rotation direction in the reward, which performed better but resulted in a back-and-forth motion.

### 9.6 MORPHOLOGY GRID SAMPLING

By sampling simulation physical parameters of the LEAP hand we find that relative morphology matter significantly. We suspect that damping and dynamic friction also play a large role but become harder to control with standard control rewards and have difficulty in correctly modeling soft bodies. These two areas remain open for further research.

## 10 REPRODUCIBILITY STATEMENT

All work can be reproduced by utilizing the open sourced code and designs. Current hardware is fully 3D printable and all electrical equipment including Dynamixel motors which have become

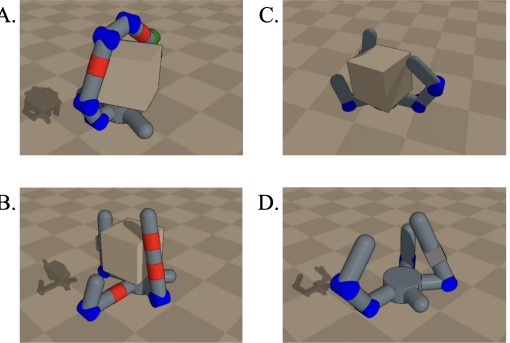

Figure 15: Design generated using RoboGrammar. A. RoboGrammar with direction-aware reward, B. RoboGrammar without directional information, C. RoboGrammar with Monte Carlo Tree Search and direction-aware reward, D. RoboGrammar with Monte Carlo Tree Search and without directional information.

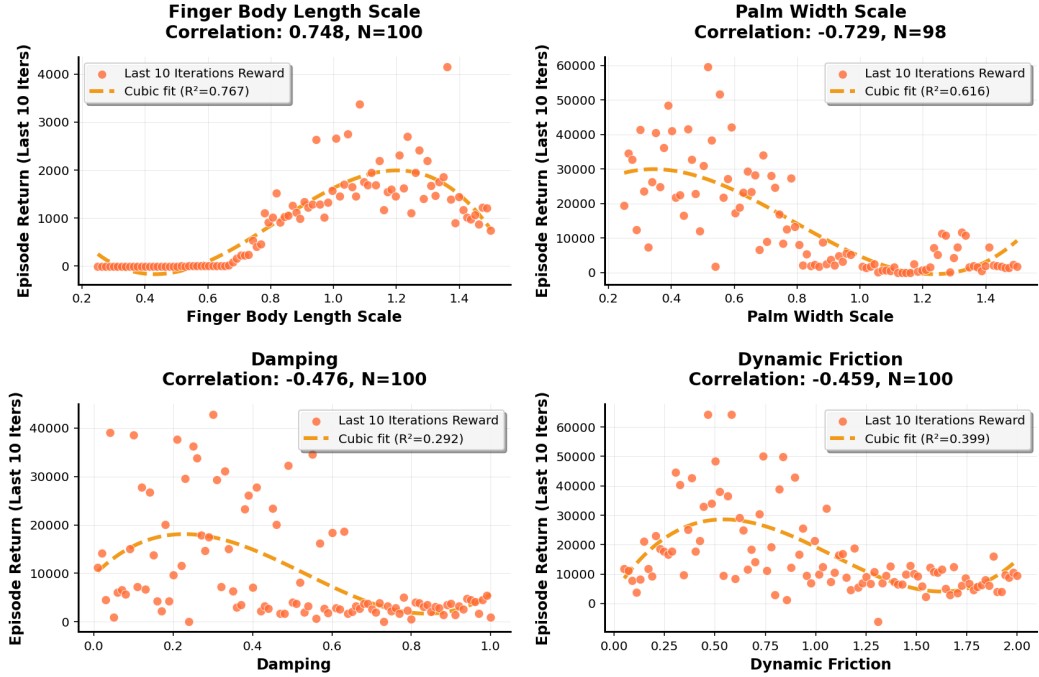

Figure 16: Top four highest parameters of impact with 100 samples completed for each parameter. For each sample, a PPO policy is retrained with an updated LEAP hand for the new hardware parameter. This is then tested for for in-hand rotation.

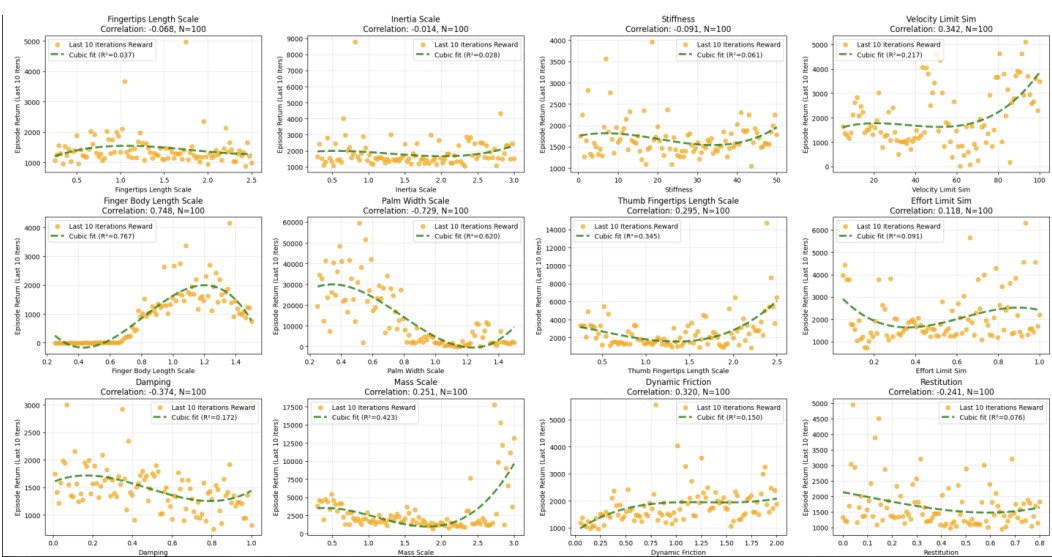

Figure 17: All sampled parameters using Bayesian sampling on the LEAP robot hand and found correlation to rotation reward.

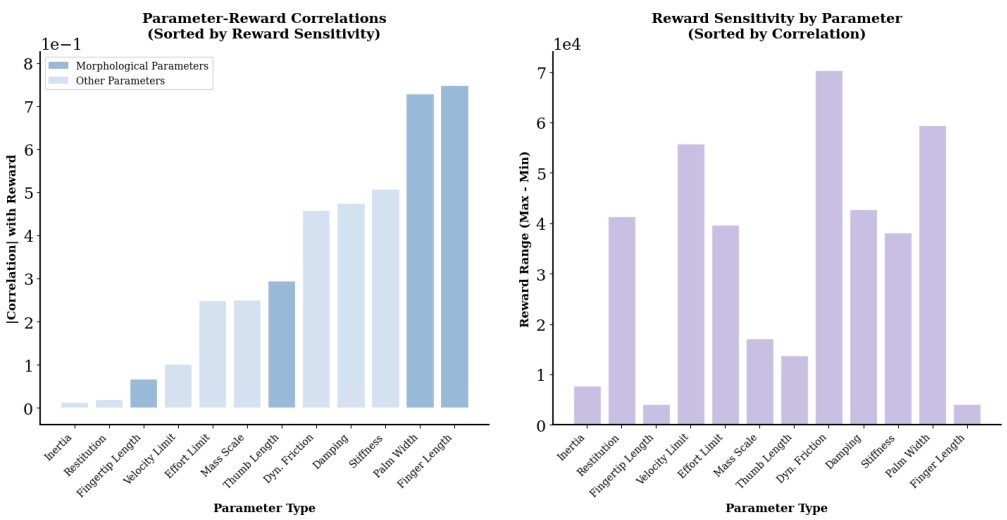

Figure 18: Ranking of each found parameter on reward impact and sensitivity of each parameter to change in reward. Results from Bayesian sampling.

Table 5: Adaptive Domain Randomization Parameters

| Parameter | Range/Value |
|---|---|
| **Object Randomization** | |
| Mass scaling | 0.9–1.3× |
| Size scaling | 0.7–1.3× |
| Position noise | 0.0–0.01 m |
| Rotation noise | 0.0–0.1 rad |
| External force acceleration | 0.5–5.0 m/s² |
| **Robot Randomization** | |
| Joint position noise | 0.0–0.05 rad |
| Joint velocity noise | 0.0–0.01 rad/s |
| Action noise | 0.1–0.2 |
| Action latency | 0–3 timesteps |
| **ADR Progression** | |
| Number of increments | 25 |
| Minimum rotation coefficient | 0.15 |
| Minimum steps for change | 960 |
| Starting increments | 0 |

increasingly utilized by labs for under $1,500 USD. Code for training existing tasks is documented alongside control for real world deployment. The authors have included instructions for adding new environments for further testing generalization capabilities, adding variable hardware and morphology constraints, and overcoming common sim-to-real issues. These efforts allow the work to be reproduced in any lab with a 3d printer and one stereo camera. Experiments can be reproduced for any individual task.

## 11 ETHICS STATEMENT

Our work completes sim-to-real co-design generation for real world robot manipulators. We hope through these efforts that positive impact such as designing more capable prosthetic hands for human neural control, pushing the understanding of tradeoffs in robot hardware, and achieving higher dexterity through co-design can be effectively explored. We acknowledge the development of more complex dexterous robot hands may enable effective automation of existing roles. We ask this research to be used for only academic and open source purposes and for users to utilize this framework as a method to evaluate tradeoffs in design and control decisions. Materials used for manufacturing primarily relied on PLA which is not widely recyclable except under industrial compostable conditions. This work significantly reduces required compute for co-design reducing training from over 100 days worth of single compute time using individual PPO policy for each design to 1.75 kw hours on a RTX 3090. This work utilizes approximately 0.6 kg of $CO_2$ which is equivalent to approximately 56 smartphone charges.

## 12 LLM ACKNOWLEDGMENT

A large language model, specifically ChatGPT 4.0 was used in polishing the writing of this paper. In particular, spelling, typos, and grammar structure were aided by GPT through all sections of the written manuscript. The authors wrote the first drafts without the use of GPT and only in final stages used GPT for polishing. Additionally, GPT was used to clean and polish our code base in the supplementary materials. This aided in formatting and removing deprecated code.

---

**Algorithm 1** Cross-Embodiment Graph Heuristic Search

---

**Require:** $N, K, \{\Theta_u\}_{u \in \mathcal{H}}, \tau_0, \Delta\tau, S, B, \rho$
**Ensure:** $G^\star, R^\star$
1: Initialize $V_\phi, \mathcal{T}[(\cdot, \cdot)] \leftarrow -\infty, \mathcal{D} \leftarrow \emptyset, G^\star \leftarrow \emptyset, R^\star \leftarrow -\infty$
2: **for** $i = 1$ to $N$ **do**
3:     $u_i \leftarrow \text{CURRENTGROUP}(i), \mathcal{C}_i \leftarrow \emptyset$
4:     **for** $k = 1$ to $K$ **do**
5:         $\tau_k \leftarrow \tau_0 + (k-1)\Delta\tau, G \leftarrow \text{GETINITIALDESIGN}(u_i)$
6:         **while** $\neg\text{ISCOMPLETE}(G)$ **do**
7:             $\Omega \leftarrow \text{OPTIONS}(G)$
8:             $o^\star \leftarrow \arg\max_{o \in \Omega} \left[ V_\phi(y(\text{APPLYRULE}(G, o)), u_i) + \tau_k\xi_o \right], \xi_o \sim \text{Gumbel}(0, 1)$
9:             $G \leftarrow \text{APPLYRULE}(G, o^\star)$
10:         **end while**
11:         $k_G \leftarrow \text{EFFECTIVEKEY}(G, u_i)$
12:         **if** $k_G$ is unique in $\mathcal{C}_i$ **then**
13:             $\mathcal{C}_i \leftarrow \mathcal{C}_i \cup \{\text{CANONICALIZE}(G)\}$
14:         **end if**
15:     **end for**
16:     $\mathcal{R}_i \leftarrow \text{CROSSEMBODIEDEVALUATE}(\mathcal{C}_i, \Theta_{u_i})$ {evaluation done in parallel}
17:     **for all** $(G_j, r_j) \in (\mathcal{C}_i, \mathcal{R}_i)$ **do**
18:         $k_j \leftarrow \text{EFFECTIVEKEY}(G_j, u_i), \ \mathcal{T}[k_j, u_i] \leftarrow \max(\mathcal{T}[k_j, u_i], r_j)$
19:         $\mathcal{D} \leftarrow \mathcal{D} \cup \{(G_j, u_i, r_j)\}$
20:         **for all** $A \in \text{PARTIALANCESTORS}(G_j)$ **do**
21:             $k_A \leftarrow \text{EFFECTIVEKEY}(A, u_i), \ \mathcal{T}[k_A, u_i] \leftarrow \max(\mathcal{T}[k_A, u_i], r_j)$
22:             with prob. $\rho$, add $(A, u_i, r_j)$ to $\mathcal{D}$
23:         **end for**
24:         **if** $r_j > R^\star$ **then**
25:             $G^\star \leftarrow G_j, \ R^\star \leftarrow r_j$
26:         **end if**
27:     **end for**
28:     **if** $|\mathcal{D}| \geq B$ **then**
29:         **for** $s = 1$ to $S$ **do**
30:             Sample minibatch $\mathcal{B} \subset \mathcal{D}, |\mathcal{B}| = B$
31:             Update $\phi$ on $\sum_{(G,u,r) \in \mathcal{B}} (V_\phi(y(G), u) - \mathcal{T}[\text{EFFECTIVEKEY}(G, u), u])^2$
32:         **end for**
33:     **end if**
34: **end for**
35: **return** $G^\star, R^\star$

---

