# OpenReview forum: "House Of Dextra : Cross-Embodied Co-Design for Dexterous Hands"
_ICLR.cc/2026/Conference — ICLR 2026 Poster_

### Official Review · Reviewer_KFiu · 2025-10-27

**Soundness:** 3
**Presentation:** 2
**Contribution:** 2
**Rating:** 4
**Confidence:** 4

**Summary:**

The paper explores the problem of the morphology-policy co-design in dexterous in-hand rotation. The problem is formulated as a bi-level optimization problem where the optimization space includes both the hand morphology and the control policy. To solve the problem, a cross-embodiment control policy is first pre-trained based on sampled morphology designs. After that, the pre-trained policy is leveraged to provide feedback for sampled designs. The cross-embodiment policy is further fine-tuned based on morphology samples. Iterating the co-design and evaluation loop helps converge on the final outcome.

**Strengths:**

- Well-motivated problem. How to increase the optimization space from pure control policies to both the control policy and the robot morphology is an important problem in robot learning. The paper is targeted at the in-hand object rotation task and proposes to design both the hand morphology and the policy.
- Reasonable methodology. Bi-level optimization is a reasonable formulation for the co-design problem. Iterating between the policy optimization and the morphology design space searching is a reasonable solution for the nested problem.

- Compared to prior co-design works, this one conducts real-world experiments and evaluations, which increases the value in real-world applicability.

**Weaknesses:**

- Spurious effectiveness. Object-in-hand rotation is a widely explored problem in the dexterous manipulation community. This work presents a research direction where the control policy and the hand morphology are co-designed. However, how does its effectiveness compare to prior in-hand rotation works, like Hora or Leap hand rotation (from Kenny Shaw). I cannot see improvements solely from the results achieved by this paper compared to prior in-hand rotation works.
- Limited analysis of the design space. The paper lacks a thorough in-depth analysis of the morphology design space, e.g., using what designs lead to satisfactory performance, and using what designs is not sufficient for a good optimization?
- Unclear methodology. How do authors iterate between the morphology design space optimization and the cross-embodiment policy training? It seems that section 3.1 does not contain this part.

Overall, this work explores an interesting direction. However, the insufficient comparisons and unclear parts in the methodology make its effectiveness and its positioning in the literature quite spurious.

**Questions:**

- Why do real-world videos only present three-fingered hands and five-fingered hands? What does this video attempt to convey? -- It seems that in some cases, the hand fails to rotate the object.
- What about the performance of this baseline -- using the leap hand rotation codebase to train a policy for each specific searched hand morphology?

---

> ### Author Response · Authors · 2025-11-25
> **Thank You!**
>
> Thank you for your interest in our work!
>
> We wanted to answer your questions and have significantly cleaned both the paper and the video to better present and label our results.
>
> ---
>
> **Q0:** Why do real-world videos only present three-fingered hands and five-fingered hands?
>
> **A0:** We have added two additional co-designed robot hands sim-to-real to show the ability of our approach to scale to other designs sim-to-real. This provides a total of three co-designed robot hands and one baseline robot hand, chosen from the random generated dataset to show tradeoff to anthropomorphic design morphology.
>
> These policies are completely blind, with no object state, no information on object type or size, and no camera is used in the system. The only input to the policy is the encoded morphology and encoders for each servo motor. The results for the four robot hands can be found here: [video](https://iclr2026-codesign.github.io/anonymous-video/).
>
> In the initial video, the top hand was the framework's best co-designed robot hand in simulation which successfully achieves blind object rotation on unseen objects without knowing object position, size, or type in real. The bottom result is the anthropomorphic robot hand baseline to show how much co-design can improve task success over traditional approaches which assume anthropomorphic hands. The best predicted co-design robot hand from the framework (3 finger hand) successfully rotates all objects shown, with no failure cases including irregular shaped objects far outside of the training dataset such as pine cones. We test on a wide range of diverse objects with our lightest object at 8.2g and heaviest object at 122g. The 18 objects have a wide variety of textures, shapes, and masses. All objects tested were unseen in training.
>
> There is no camera in this setup and no tactile sensing, the policies rely only on encoder information as the hand manipulates the object. This makes the task more difficult as the robot hands need to accurately infer object state from position error across each motor alone - hence the example of failure cases for the baseline hands.
>
> ---
>
> **Q1:** What about the performance of this baseline -- using the leap hand rotation codebase to train a policy for each specific searched hand morphology?
>
> **A1:** We added a comparison of LEAP rotation tasks (with and without object state) to understand performance trade offs. Results for LEAP from the in-hand rotation in Isaaclab for 5000 epochs are reported below. In simulation, our design performs over a 4× performance to LEAP with vision. Additionally, when we added our training objects to the reorientation codebase, the policy failed to generalize to many objects - even after increasing our objects by 20% to account for LEAP's larger size and increasing epochs. The reorientation codebase was fine tuned for a cube, so we report that result for the best comparison.
>
> | Table | Blind | Vision |
> |-------|-------|--------|
> | 3 Finger Co Design | 1.8 rad/sec | 3.3 rad/sec |
> | Anthro Baseline | 0.56 rad/sec | 0.73 rad/sec |
> | LEAP, cube | 0.0 rad/sec | 0.47 rad/sec |
>
> We think this low performance for blind rotation for LEAP is likely a control issue. Cross embodied training significantly outperforms an individual PPO policy across designs we test. Training the single five finger symmetric from scratch, PPO on a single robot hand achieves 0.56 rad/sec of rotation. By contrast, the cross embodied policy deployed with no fine tuning on the same morphology achieves a 65% improvement. The cross embodied policy also has less failure across objects while PPO trained for a single morphology fails, achieving only minimal rotation, on the same training object set. Results are shown in the updated video.
>
> In terms of hardware comparison, our largest robot hand is ~10% smaller in length than LEAP and has 5 fingers while LEAP only has 4. We created our baseline in order to better benchmark anthropomorphic morphology of 5 fingers. It is also fully modular for mechanical and electrical parts, allowing us to create the other three robot hands from its electrical and newly printed mechanical parts. We have added this to our contributions as this platform can be used easily for further co-design research in manipulation.
>
> ---
>
> **Q2:** Limited analysis of the design space. The paper lacks a thorough in-depth analysis of the morphology design space, e.g., using what designs lead to satisfactory performance, and using what designs is not sufficient for a good optimization?
>
> **A2:** We have updated our methodology to include details in generation and space optimization. Please see section 3.2 for added description of design generation and design optimization.
>
> ---
>
> We thank the reviewer for their feedback. Thank you!

---

### Official Review · Reviewer_LTDD · 2025-10-29

**Soundness:** 2
**Presentation:** 2
**Contribution:** 3
**Rating:** 4
**Confidence:** 4

**Summary:**

This paper aims to address the challenge of co-designing dexterous robotic hands, where both the physical morphology and the control policy are optimized simultaneously. The authors leverage a cross-embodiment control policy to overcome the computational intractability of this problem. This universal policy is pre-trained on a diverse set of robot hand designs generated by a procedural grammar. Morphologies are encoded using GNNs to condition the policy, and a learned design value network guides the search for optimal designs. The framework is validated on several simulated manipulation tasks and through the fabrication and zero-shot deployment of an optimized 3-fingered hand.

**Strengths:**

- The paper’s strongest aspect is its rigorous validation on a full-stack physical platform from hardware design to policy deployment.
- The proposed method directly tackles the computational cost of searching the joint design-control space in co-design tasks. The framework's total design-to-deployment time of under 24 hours is a significant engineering achievement and a key selling point for its real-world utility.
- The work provides valuable insights that extend beyond the specific designs found, contributing to the broader understanding of dexterous manipulation.

**Weaknesses:**

- The bi-level optimization problem in Eqs. (2)-(4) implies a joint optimization, but the described method involves first pre-training a general policy and then using it for evaluation. The connection between the formal problem and the practical algorithm is not well-established, making the algorithm feel more like an effective heuristic than a direct solution to the stated objective.
- The real-world validation does not compare the co-designed system against a sufficiently strong, specialized baseline, making claims of overall system superiority difficult to substantiate. The experiment compares the optimized 3-fingered hand and the baseline anthropomorphic hand by deploying the same morphology-conditioned cross-embodiment policy on both. This effectively tests which morphology performs better given the proposed adaptive control framework. This comparison does not rule out the possibility that a state-of-the-art policy, trained specifically for the anthropomorphic hand, could outperform the co-designed system. The anthropomorphic hand's failure could stem from the universal policy's inability to master its higher-DOF control space, rather than a fundamental flaw in the hand's design for the task. To make a stronger claim about the superiority of the final co-designed system, the baseline should be a strong, existing design/policy pair (e.g., an anthropomorphic hand with a dedicated, highly-tuned policy), not just a standard design controlled by the paper's own universal policy.
- Table 1 should be split into two tables. It combines two conceptually separate tables into a single, side-by-side layout under one caption. It is confusing whether the rows are meant to be read horizontally across the entire table or as two independent blocks.

**Questions:**

See weaknesses.

---

> ### Author Response · Authors · 2025-11-25
> **Thank you!**
>
> We thank you for your thoughtful analysis and for your time.
>
> ---
> **Q0:** The bi-level optimization problem in Eqs. (2)-(4) implies a joint optimization, but the described method involves first pre-training a general policy and then using it for evaluation.
>
> **A0:** Thank you for this thoughtful question. We would like to clarify that Eq 2-4 are not meant to represent our algorithm, but rather to define the goal of the co-design problem: the joint optimization objective over morphology and control. While this cross-embodied policy serves as a form of heuristic, it is not an ad hoc heuristic but a policy-grounded approximation of the actual controller, trained through PPO across a diverse distribution of designs. It effectively captures how control should adapt to morphology and serves as a reusable backbone for efficient design evaluations. We have modified the paper accordingly to clarify.
>
> ---
> **Q1:** The real-world validation does not compare the co-designed system against a sufficiently strong, specialized baseline, making claims of overall system superiority difficult to substantiate.
>
> **A1:** To help find a suitable baseline, we compared a blind leap hand policy to both our initial baseline and the 3 finger co-design hand in simulation with fine tuned PPO from the LEAP-Hand_Reorient code base. By contrast, our worst sim-to-real hand (anthropomorphic baseline) achieves ~20% improvement over LEAP - partially due to having five fingers while LEAP only has four. The co-design three finger robot hand deployed with the cross embodied policy without any fine tuning achieves a 4× improvement compared to LEAP fine tuned.
>
> The LEAP test used the PPO policy from LEAP hand's [reorientation task](https://github.com/leap-hand/LEAP_Hand_Isaac_Lab). In order to test how their policy performs blind (as our results sim-to-real are blind with no object state), we completed another test training from scratch with a blind policy. However, the LEAP hand had no clear rotation and failed to generalize over our test objects, even when increasing run time and object size. Our original baseline anthropomorphic robot hand has five fingers (while LEAP has only four) and is 10% smaller in length than LEAP with no loss in degrees of freedom per finger. This helps make our baseline a more challenging morphology to beat for rotation.
>
> - 3 Finger Hand w/o fine tuning + Cross Embodied Policy: 1.85 rad/sec
> - Anthropomorphic Baseline w/o fine tuning + Cross Embodied Policy: 0.56 rad/sec
> - LEAP Hand Reorientation w/ PPO: 0.47 rad/sec
>
> We ask the reviewer to consider that the primary focus of our paper is not to outperform manually designed robot hands or fine tuned control for rotation, but to show a tractable co-design method with strong performance on complex manipulation tasks (flipping, rotation, grasping) and one task sim-to-real. Limitations prior to our work include lack of scalable policy evaluation, understanding parameters of impact, and lack of suitable hardware platform for deploying multiple designs sim-to-real. We agree strong baselines are important, but ask to be evaluated in the context of broader co-design work.
>
> We added two more co-designed robot hands deployed sim-to-real (blind) to help show hardware scalability of our modular robot hands and impact of design on task success. The updated video to show these additional results sim-to-real can be found in the updated video.
>
> **1/2**

---

> > ### Author Response · Authors · 2025-11-25
> > **Official Comment by Authors**
> >
> > **Q2:** Table 1 should be split
> >
> > **A2:** The tables are split! We have also updated the overall work for clarity and presentation.
> >
> > **Q3:** The connection between the formal problem and the practical algorithm is not well-established, making the algorithm feel more like an effective heuristic than a direct solution to the stated objective.
> >
> > **A3:** To help resolve the bi-level optimization problem, we tested our cross embodied policy performance without fine tuning compared to the policy checkpoint fine tuned for the sampled design. We also compare PPO to training from scratch on a single design, to show the advantages of a cross-embodied policy performance and better generalization across objects from a more diverse dataset.
> >
> > Cross embodied training significantly outperforms an individual PPO policy. Training the single five finger symmetric hand from scratch, PPO on a single robot hand achieves 0.56 rad/sec of rotation. By contrast, the cross embodied policy deployed with no fine tuning on the same morphology achieves a 65% improvement. The cross embodied policy also has less failure across objects while PPO trained for a single morphology fails, achieving only minimal rotation, on the same object set. Results are shown in the updated [video](https://iclr2026-codesign.github.io/anonymous-video/).
> >
> > For the bi-level optimization problem, we found that the rate of improvement across designs with and without fine tuning was similar. For example, the symmetric five finger robot hand achieves 0.93 rad/sec without fine tuning, and 1.87 rad/sec with fine tuning. Likewise, the three finger robot hand achieves 1.69 rad/sec without fine tuning and 3.3 rad/sec with fine tuning. This allows us to evaluate best control-design pairs, and then fine tune the best found design as the ranking of the designs remains consistent. Fine tuning takes a significant amount of added time (~40 minutes per design or more). Due to this, we viewed the tradeoff in time for fine tuning all designs as not worth completing for all designs.
> >
> > --
> >
> > With the addition of the two additional sim-to-real robot hands, baseline comparison to leap, and expanded results of the three tasks, we hope these additions help significantly improve our paper.
> >
> > Please let us know if the baselines are suitable or if additional analysis is needed. Thank you!
> >
> > **2/2**

---

### Official Review · Reviewer_hBj2 · 2025-11-01

**Soundness:** 2
**Presentation:** 3
**Contribution:** 3
**Rating:** 4
**Confidence:** 2

**Summary:**

The authors introduce a co-design framework that jointly optimizes the control policy and dexterous hand morphology given a specific task. This problem is high dimensionality in both the policy and manipulator space. The authors claim that by grouping manipulator morphologies  into classes they address this issue. As far as I can tell this has not been shown empirically or theoretically. However, the experimental analysis does indicate the utility of the co-design strategy. For example, different manipulator morphologies are identified for in hand rotation and flipping.

**Strengths:**

**Originality.** The work appears original.

**Quality.** The work appears to be of good quality. The work is well motivated (co-design is very high-dimensional), but it is not clear to me how the approach addresses the challenge indicated in the motivation.

**Clarity.** The paper is well written and well organized.

**Significance.** The significance of co-design is clear, and the authors demonstrate that the proposed approach is an effective strategy for co-design.

**Weaknesses:**

My main uncertainty about this paper is whether the claim that the proposed approach reduces the dimensionality of the problem has been shown.

**Questions:**

**Questions/Comments**
- Line 096: what are X_v and X_e here? The feature on each node and edge?
- As I understand, a GNN wouldn’t be able to differentiate between a hands with finger arrangements related by permutation. This seems undesirable since a radially symmetric five fingered hand with two shorter fingers next to each other would likely benefit from a different control policy than a hand with two shorter fingers separated by two longer ones.

**Possible typos**
- Line 269: These tasks comprise → These tasks include
- Table 1: It would be a little easier to read if the strongest performing results were in bold
- Table 1: It seems like this should be two separate tables?
- Line 300: Robogrammar achieves an average rotation speed of .367… This value is different than what is reported in Table 1
- Line 320: anc → and
- Line 321: radial symmetrical → radial symmetry
- Line 324: 1.7 radians/sec → 1.7 rad/s. (so the notation is consistent)
- Line 354: secventeen → seventeen
- Line 137: G\in \mathcal{D} → G\in \mathcal{G}

---

> ### Author Response · Authors · 2025-11-25
> **Thank You!**
>
> We thank the reviewer for their insightful questions and for recognizing that our work is well-motivated and of good quality. Co-design is high-dimensional and we cannot change the actual dimensionality of this problem. Instead, we aim to make this problem more tractable. Our method of using a cross-embodied control policy serves as an effective way to explore this large joint space. With this adaptive control backbone, we can evaluate designs efficiently without the need to recompute control policies for individual designs. We have modified the paper accordingly to clarify this.
>
> To help prove tractability, we compared the run time for the cross embodied policy to individual PPO policies. Using PPO to train each design took over 26 hours, and only 20 designs were able to be evaluated with over 45 minutes needed per design. By contrast, our cross embodied evaluation evaluated 2,000 designs in 5.18 hours without fine tuning. For fair PPO comparison, we lowered the epochs to what we observed was an individual hand's time to convergence (350 epochs, instead of 1000 for the crossed embodied policy). This ensures a more fair comparison of runtime to the baseline.
>
> **Our cross embodied evaluation is over 400× faster in run time than PPO and can evaluate and generate and evaluate 1,800 more designs in ⅕ of the run time.**
>
> **Cross embodied training also significantly outperforms an individual PPO policy.** For example, training the single five finger symmetric from scratch, PPO on a single robot hand achieves 0.56 rad/sec of rotation. By contrast, **the cross embodied policy deployed with no fine tuning on the same morphology achieves a 65% improvement.** The cross embodied policy also has less failure across objects while PPO trained for a single morphology fails, achieving only minimal rotation, on the same training object set. Results are shown in the updated [video](https://iclr2026-codesign.github.io/anonymous-video/).
>
> Additionally, we shrunk the design space of co-design by finding parameters of impact using Bayesian Sampling [Why Morphology? Section] - such as identifying that material properties were not as high impact. This included reducing hardware parameters to morphology and control only and helped motivate our use of co-design which handles both. Material properties such as compliance, friction, or stiffness were removed from the search space due to their low impact.
>
> Lastly, by creating modular components, this allowed robot hands to be co-designed sim-to-real across designs. To help show the tractability of modular hardware, we also added two additional co-designed robot hands sim-to-real. These hands are built from the same modular parts allowing easy transfer to different designs and tractable sim-to-real by using the same actuators. Results are added in the paper.
>
> ---
>
> **Q0:** Line 096: what are $X_v$ and $X_e$ here? The feature on each node and edge?
>
> **A0:** Yes, the feature on each node and edge.
>
> ---
>
> **Q1:** As I understand, a GNN wouldn't be able to differentiate between a hands with finger arrangements related by permutation. This seems undesirable since a radially symmetric five fingered hand with two shorter fingers next to each other would likely benefit from a different control policy than a hand with two shorter fingers separated by two longer ones.
>
> **A1:** We can! We use a design encoder (using one hot encoding) before giving this to the GNN. This allows us to distinguish between permutations such as 5 fingered anthropomorphic hands and 5 fingered symmetric hands.
>
> Using your example, suppose two radially symmetric 5 fingered hands - one with two shorter fingertips next to each other, and the other spaced apart. In the example of the five fingers with thin fingertips spaced apart fingers the encoding would look something like this:
> ```
> Node 1: [..., position one-hot: [0, 1, 0, 0, 0, 0], tip: [0, 1, 0], ...]
> Node 2: [..., position one-hot: [0, 0, 1, 0, 0, 0], tip: [1, 0, 0], ...]
> Node 3: [..., position one-hot: [0, 0, 0, 1, 0, 0], tip: [1, 0, 0], ...]
> Node 4: [..., position one-hot: [0, 0, 0, 0, 1, 0], tip: [1, 0, 0], ...]
> Node 5: [..., position one-hot: [0, 0, 0, 0, 0, 1], tip: [0, 1, 0], ...]
> ```
>
> While for adjacent thinner fingers the encoding would be:
> ```
> Node 1: [..., position one-hot: [0, 1, 0, 0, 0, 0], tip: [0, 1, 0], ...]
> Node 2: [..., position one-hot: [0, 0, 1, 0, 0, 0], tip: [0, 1, 0], ...]
> Node 3: [..., position one-hot: [0, 0, 0, 1, 0, 0], tip: [1, 0, 0], ...]
> Node 4: [..., position one-hot: [0, 0, 0, 0, 1, 0], tip: [1, 0, 0], ...]
> Node 5: [..., position one-hot: [0, 0, 0, 0, 0, 1], tip: [1, 0, 0], ...]
> ```
>
> The GNN can detect the difference through position one-hot encoding, combined with an indicator for fingertip type.
>
> ---
>
> Please let us know if there are additional concerns. Thank you!

---

### Official Review · Reviewer_P9ND · 2025-11-02

**Soundness:** 3
**Presentation:** 3
**Contribution:** 3
**Rating:** 6
**Confidence:** 4

**Summary:**

This paper presents a co-design framework that jointly optimizes dexterous hand morphology and control policies for in-hand rotation tasks. The approach uses morphology-conditioned policies, graph neural networks for design encoding, modular hardware grammars and graph heuristics search. The framework is evaluated in both simulation and the real world, demonstrating that optimized designs outperform baselines.

**Strengths:**

- The use of cross-embodiment control policy for task-specific hardware design is well-motivated.
- Experiments are done both in sim and real, especially with solid real world results.

**Weaknesses:**

- The task is a bit limited and narrow - focusing solely on in-hand rotation.
- The method uses a search-based approach to find optimal designs, which is very slow. Also the graph heuristic search explores only 50 iterations × 25 designs = 1,250 total designs, which seems limited given the design space complexity.

**Questions:**

- The run time reported in table 1 (i assume is the time for each approach to search for one design) is very slow. Have the authors considered using optimization methods (zero order such as CMA-ES, bayesian optimization and first order such as gradient descent, L-BFGS, diffusion guidance) rather than search based algorithms?

---

> ### Author Response · Authors · 2025-11-25
> **Thank you!**
>
> We thank the reviewer for identifying our contributions to co-design for in hand rotation including a cross embodied control policy for efficient search and compelling sim to real results.
>
> ---
>
> **Q0: The task is a bit limited and narrow - focusing solely on in-hand rotation.**
>
> **A0:** We expand our results to three tasks by including further results on the Grasping and Flipping Tasks. These two tasks are applied across randomized objects, showing the ability of our method to generalize to a variety of manipulation skills. Visualization of these results in simulation can be seen in Fig. 5, metrics in our results section, and in our updated [__video__](https://iclr2026-codesign.github.io/anonymous-video/).
>
> We primarily focused on blind in-hand rotation due to its difficulty, but have expanded our results to showcase our other two tasks. Additionally, we expand our sim-to-real results by adding two more co-design robot hands sim-to-real.
>
> ---
>
> **Q1: The run time reported in table 1 (i assume is the time for each approach to search for one design) is very slow. Have the authors considered using optimization methods (zero order such as CMA-ES, bayesian optimization and first order such as gradient descent, L-BFGS, diffusion guidance) rather than search based algorithms?**
>
> **A1:** The run time reported is for generating and evaluating all 2,000 designs in simulation per task. The result for each task is a ranking of each found design and an adaptive cross-embodied policy for each morphology.
>
> We thank the reviewer for this insight on using zero order methods. On the first attempt at solving this problem, we used bayesian optimization to sample design parameters but found rewards sensitive to small changes in design parameters and Bayesian sampling needed many samples. Bayesian methods also still needed a control policy for evaluation after resampling parameters, meaning PPO would need to be rerun for each design - hence the tractability issue we addressed with cross embodiment.
>
> For first order approaches, gradient descent often behaves poorly on complex geometry [[__Neural Graph Evolution__](https://arxiv.org/abs/1906.05370)], diffusion guidance struggles in structure and guaranteeing physically realistic constraints are met [[__DiffuseBot__](https://diffusebot.github.io/)]. These methods are also sensitive to initialization, so a very good guess for initializing the network is needed. Other methods are also slow for evaluation, which we aim to improve through cross embodiment pre-training.
>
> ---
>
> **Q2: The method uses a search-based approach to find optimal designs, which is very slow. Also the graph heuristic search explores only 50 iterations × 25 designs = 1,250 total designs, which seems limited given the design space complexity.**
>
> **A2:** We have updated our results for running 50 × 40 designs, as we find the performance does not increase significantly beyond 1,500 designs (Fig. 5) across the three tasks we created. Graph Heuristic Search learns an epsilon greedy strategy to predict which design types perform best and trades off between random search and exploring branches of similar designs to those that are well performing. This allows the search to be efficient enough across a smaller search horizon. We have also made small changes to our code to parallelize environments more efficiently at evaluation. The run time for generating and evaluating 2,000 designs is now 5.18 hours without fine tuning for the best found design.
>
> ---
>
> We thank the reviewer for their insights. We believe that our additional experimental results and response address your concerns. However, please do not hesitate to let us know if you have additional comments.

---

### Author Response · Authors · 2025-11-25
**General Comment and Overview of Changes**

We thank all the reviewers for their thoughtful insights and valuable feedback, and have incorporated their suggestions into the revised manuscript; the list of changes are outlined below. We have also responded to your individual comments.

**Summary of revisions:** We have made strong revisions to our manuscript to meet your concerns, with the list of these key revisions below:

* **(All)** Our [video](https://iclr2026-codesign.github.io/anonymous-video/) is updated for additional sim-to-real results and visual simulation results of the three dexterous tasks: grasping, flipping, and in-hand rotation. We have added the modular robot hand platform to part of our key contributions due to being able to quickly create high-performance robot hands for the research community.

* **(KFiu)** We have added results for two additional co-design robot hands sim-to-real for the in-hand rotation task to show the ability of our algorithm to accurately predict control-design ranking sim-to-real - with the optimal found design, 3 finger, being the best with no object state. In total, four co-design robot hands are now deployed sim-to-real.

* **(hBj2)** We provide an initial comparison to PPO results in Appendix A, showing our policy is 400x faster without loss of dexterity. Cross embodiment also achieved better generalization across objects and rotation speed compared to PPO trained from scratch.

* **(LTDD, KFiu)** We show our work is 4x faster than LEAP for the fine tuned PPO policy on the LEAP-Reorient-Task for in-hand-rotation with object state. We test LEAP with vision, without vision, and the same corresponding objects to our general object dataset. Without object state, LEAP provides no rotation in simulation.

* **(P9ND)** We have revised our paper to more clearly showcase our three tasks in simulation: grasping, flipping along the z axis, and in-hand rotation. We provided additional results for grasping and flipping to show best found designs and task performance.

* **(P9ND)** We also show by running for 2,000 designs for each task (Fig. 5) that performance does not increase substantially after 1,500 generated designs.

* **(KFui)** We add design optimization iteration and generation to our methodology section. Additionally, we have expanded our design analysis for each task.

* **(LTDD)** We show that fine-tuned policies result in similar scale of rotation speeds.

We have also improved the presentation and clarity of our work by remodeling our figures for clarity. Note, we have also made a number of small changes to the paper for writing clarity and to correct grammatical errors.

Best,
*Authors of Cross-Embodied Co-Design for Dexterous Hands*

---

### Meta-Review · Area_Chair_HvhW · 2026-01-02

**Summary:**

This paper explores cross-embodied co-design of dexterous hand morphology and control via a morphology-conditioned universal policy. Reviewers point out limitations in task coverage, baseline strength, and experimental depth, but also recognize the coherence of the framework and the successful sim-to-real deployment on physical hardware. Taking the full discussion and rebuttal into account, the AC views the work as addressing an important problem with a clear system-level contribution and recommends acceptance, despite the relatively simple experimental setting.

**Reviewer Concerns:**

Reviewer concerns mainly relate to evaluation scope and methodological depth. Several reviewers question whether the proposed bi-level formulation is explicitly optimized and whether the reported gains are competitive against more specialized controllers or broader task suites. These points are clarified in the rebuttal through additional experiments and a clearer positioning of the method as a practical co-design framework rather than a theoretically optimal solver. While the experimental coverage remains limited, the concerns are not viewed as blocking, and the work’s contribution is considered adequate given its focus and scope.

**Reviewer Scores:**

- Reviewer P9ND: Scores the paper 6, recognizing solid engineering and sim-to-real validation but noting narrow task coverage and slow search. After the rebuttal, the assessment would likely remain borderline (around 6).
- Reviewer hBj2: Scores the paper 4, questioning whether the approach meaningfully reduces co-design complexity and whether key claims are demonstrated. These concerns persist after rebuttal, and the score would likely remain unchanged (4).
- Reviewer LTDD: Scores the paper 4, citing weak baselines and unclear comparisons. After the rebuttal addresses these issues, the assessment would move to a borderline level (6).
- Reviewer KFiu: Scores the paper 4, recognizing the relevance of the problem but questioning experimental depth. After the rebuttal and discussion, the assessment would increase to a borderline level (6).

---

### Decision · Program_Chairs · 2026-01-26

Accept (Poster)